# Environmental Influences of High-Density Agricultural Animal Operation on Human Forearm Skin Microflora

**DOI:** 10.3390/microorganisms8101481

**Published:** 2020-09-26

**Authors:** Mengfei Peng, Debabrata Biswas

**Affiliations:** 1Department of Animal and Avian Sciences, University of Maryland, College Park, MD 20742, USA; murphy7@umd.edu; 2Center for Food Safety and Security Systems, University of Maryland, College Park, MD 20742, USA

**Keywords:** human skin microbiome, farm animal, dysbiosis, microbial diversity, metagenomics

## Abstract

The human forearm skin microbiome ecosystem contains rich and diverse microbes, which are influenced by environmental exposures. The microbial representatives can be exchanged between human and environment, specifically animals, by which they share certain or similar epidermal microbes. Livestock and poultry are the microbial sources that are associated with the transmission of community-based pathogenic infections. Here, in this study, we proposed investigating the environmental influences introduced by livestock/poultry operations on forearm skin microflora of on-site farm workers. A total of 30 human skin swab samples were collected from 20 animal workers in dairy or integrated farms and 10 healthy volunteer controls. The skin microbiome was 16S metagenomics that were sequenced with Illumina MiSeq system. For skin microbial community analysis, the abundance of major phyla and genera as well as alpha and beta diversities were compared across groups. We identified distinctive microbial compositional patterns on skin of workers in farm with different animal commodities. Workers in integrated farms containing various animals were associated with higher abundances of epidermal Proteobacteria, especially *Pseudomonas* and *Acinetobacter*, but lower Actinobacteria, especially *Corynebacterium* and *Propionibacterium*. For those workers with frequent dairy cattle operations, their Firmicutes in the forearm skin microbiota were enriched. Furthermore, farm animal operations also reduced *Staphylococcus* and *Streptococcus*, as well as modulated the microbial biodiversity in farm workers’ skin microbiome. The alterations of forearm skin microflora in farm workers, influenced by their frequent farm animal operations, may increase their risk in skin infections with unusual pathogens and epidermal diseases.

## 1. Introduction

As the largest organ of human body, the skin plays the primary role of physical barrier defending against the penetration of exogenous microorganisms and adverse environmental conditions [1]. Meanwhile, distinctive communities of enormous microorganisms, especially bacterial commensal inhabit on the skin, forms the diverse and complex microbial ecosystem as an epidermal microfloral community [2,3] and it plays an important role for the host. These symbiotic microbes-host relation actively protects the colonization of harmful or unexpected pathogens, induces immunity, and influences lipid metabolism, body odor production, and many other beneficial activity of the human host [4,5]. However, the microbial composition and their functions are site- and condition-specific, which are driven and shaped by the local moisture, temperature, pH, sebaceous condition, and the density of hair follicles and sweat glands [1,3,4,6].

Throughout the entire human skin, the microbial community on the forearm is one of the richest and it contains the most diverse microbes specifically bacterial community, with the dominance of four phyla as Actinobacteria, Bacteroidetes, Firmicutes, and Proteobacteria [2,7]. Furthermore, several consistently abundant genera in the epidermal bacterial community that were widely studied include *Acinetobacter*, *Corynebacterium*, *Propionibacterium*, *Pseudomonas Staphylococcus*, and *Streptococcus* [7,8]. The host genotype, individual physiology, and personal lifestyle (i.e., occupation, hygiene, clothing, and medication, especially antibiotic use) contribute a significant role in developing human skin microbiome and their diversity [5]. At the same time, the constant exposure to a particular environment, for instance, the frequent contact with pets at home or farm animals in agricultural farm, is capable of re-disturbing the homeostatic skin microbiota and further establishing a divergent microbial ecosystem in the exact location and neighbors of the contact skin area [9,10].

The microbial elements in skin microflora can easily transfer and share between animals and human individuals, which is much conspicuously observed on pets and their owners [11]. A prominent similarity in skin microbiome was found among individuals living with pets in comparison with those who own no pet [12]. The particular skin microbiota was observed to be analogous among humans and dogs in a shared household [13], as well as among researchers and research animals in the laboratory [14]. As a matter of fact, the animal ownership and husbandry were significantly correlated with increased microbial diversity in the master’s skin microbiome [11,13,15].

On the other hand, the transmission of community-based infectious pathogens can occur by means of both direct and indirect contact within a certain environment [16]. Especially, epidermal wound infections are caused by bacterial pathogens, including *Staphylococcus aureus* during the disruption of skin barrier and exposure to the environment, specifically animals [16,17]. Such skin infection and pathogen transformation in community settings have been identified as the major spreading cases of community-acquired methicillin-resistant *S. aureus* (MRSA) from animal origin, during which the susceptible epidermal *Staphylococcus* species were usually replaced by MRSA inducing highly virulent infections in high-risk populations [18,19,20]. Furthermore, livestock and poultry, possessing numerous microbes, including zoonotic pathogens throughout their bodies [21,22], are broadly domesticated in different types of agricultural farms for generating food products, including meat, milk, and eggs, as well as fur and leather commodities [23]. For example, cow, goat, sheep, camel, and donkey are reared in dairy farms for milk production [24,25]; chicken, turkey, duck, and goose are grown in poultry farms for either meat or egg production [26,27]; moreover, pig, buffalo, and rabbit are raised in various integrated farms, including mixed crop-livestock farms for their meat products [28,29]. These farm animals are major reservoirs of multiple zoonotic pathogens and they cause the transmission of antibiotic-resistance bacterial pathogens [30,31,32,33]. The farm workers who have frequent operations on these farm animals are at greater risk of altering their microbiome, particularly the skin of their hand and forearm microbiome, with animal-origin bacterial pathogens possessing antibiotic resistance [11]. However, such influence brought by frequent farm animal operations on human skin and their impact on the health of these farm workers are still unclear.

This study aimed to explore the environmental influences, specifically frequent farm livestock operations, on the skin microbiome of on-site farm labors. To achieve such goals, we investigated the differential microbial structures of forearm skin microbiota among workers in distinct dairy or mixed (integration of animals and crops) farm environments, and we compared them with common individuals without farm animal contact. The potential influences on specific groups of microbes on skin microbiome that were introduced by animal operations and their connections were further discussed.

## 2. Materials and Methods

### 2.1. Sample Collection and Processing

A total of 30 individuals, including 20 farm workers of dairy or integrated farms in Maryland-Washington DC (Clarksville farm (*n* = 9), Upper Marlboro farm (*n* = 8), and Washington DC (*n* = 3)) and 10 healthy volunteers in the University of Maryland—College Park (UMCP) campus were enrolled in this study (Table 1). The farm workers were selected based on the criteria: (1) no antibiotic treatment within the recent 30 days; (2) continuously working on-site in a farm for at least 10 days or longer; (3) the work duty in the farm involving personal animal operations; and, (4) no coverage of forearms by personal protective equipment. The volunteers, mostly students, staffs, and faculties, on the UMCP campus were selected based on the criteria: (1) no antibiotic treatment within the recent 30 days; (2) did not visit any animal farm within the recent 10 days; and, (3) no personal contact with farm animals and pets within the recent 10 days. All of the skin samples were collected from the outside near-palm portion of the small arms by sterilized cotton swabs, maintained in 0.5 mL phosphate buffered saline (PBS) in Eppendorf tubes (VWR, Radnor, PA, USA), and transferred back to laboratory in an ice cooler. The skin swabs were collected twice from each participated individuals, and the duplicates were combined for sequencing and analyses. Prior approval (1206869-2) has been obtained from the Institutional Review Board, University of Maryland, for this study.

### 2.2. DNA Isolation and 16S Ribosomal RNA Gene Sequencing

The microbial genomic DNA was extracted from individual skin swab sample using the PureLink Microbiome DNA Purification Kit (Invitrogen, Carlsbad, CA, USA), following the instructions from the manufacturer. The duplicated DNA from the same participant is combined and recognized as one DNA sample. The isolated DNA samples were stored at −20 °C until further use for library preparation. The 16S rRNA variable V3 and V4 regions were targeted for amplification of microbial gene-specific sequences for next-generation sequencing-based phylogenetic classification and diversity analysis. The PCR amplification was carried out in a volume of 25 μL reaction system with 2× KAPA HiFi HotStart ReadyMix (KAPA Biosystems, Wilmington, MA, USA), 12.5 ng microbial DNA, and 1 μM primer pairs, following the program: 95 °C for 3 min., 25 cycles (95 °C for 30 s, 55 °C for 30 s, and 72 °C for 30 s), and 72 °C for 5 min. The amplicons were further cleaned and purified with AMPure XP beads (Beckman Coulter Genomics, Danvers, MA, USA), following the manufacturer’s instructions. Dual indices and adapters were linked with the amplicon using Nextera XT Index Kit v2 Set C (Illumina, San Diego, CA, USA) following the program: 95 °C for 3 min., 8 cycles (95 °C for 30 s, 55 °C for 30 s, and 72 °C for 30 s), and 72 °C for 5 min., followed by second clean-up with AMPure XP beads. The equimolar-pooled DNA libraries were prepared with Nextera XT DNA Library Preparation Kit (Illumina, San Diego, CA, USA), according to the manufacturer’s instruction. The final library containing the reference control PhiX Control v3 (Illumina, San Diego, CA, USA) and negative control (DNA extracted from blank swabs open to the same environments), after heat denaturation (96 °C for 2 min.), was loaded for 2 × 300 bp paired-end reads sequencing with Illumina MiSeq system using v3 600-cycle kit (Illumina, San Diego, CA, USA). All of the raw sequence data were submitted to GenBank SRA under BioProject PRJNA633151, accession numbers SRR11803381- SRR11803410.

### 2.3. Dataset Processing and Quality Control

BCL2FastQ and DeconSeq were applied for demultiplexing and PhiX removing, separately, from the raw sequencing data. The paired-end data consisting of separate FASTQ files were further filtered and trimmed while using mothur (version 1.44) toolsuites. Briefly, the initial clean-up was performed by Screen.seqs tool to remove problematic reads, including contigs longer than 250 bp and ambiguous bases, for reducing bias in downstream analysis. The unique reads were further trimmed with Screen.seqs tool for ensuring that all of the reads overlapped with the V3/V4 region of 16S rRNA gene. Filter.seqs tool was used for removing overhangs and gap characters, and Pre.cluster tool was applied to merge near-identical sequences with the threshold of 1% mismatches. The chimera hybrid sequences that were generated by mis-priming were then removed from the dataset by VSEARCH algorithm. Taxonomic classification was carried out by Classify.seqs tool while using SILVA reference database (version 138) based on RDP classifier algorithm (version 11) [34]. Subsequently, sequence contaminations including 16S/18S rRNA gene fragments from Archaea, chloroplasts, and mitochondria were filtered by Remove.lineage tool. The Operational Taxonomic Units (OTUs) were defined and clustered at the 97% identity threshold of the 16S rRNA gene sequence variants.

### 2.4. Analyses of Microbial Taxonomy and Diversity

The relative taxonomic abundance (RTA) of a specific taxon was normalized and calculated based on the following formula for comparison and statistical analysis: RTA = Number of Reads in taxon/Number of Reads in Total 16S rRNA gene. The difference in microfloral structures among groups was determined based on analysis of composition of microbiomes with ANCOM package in R [35]. Rarefaction.single tool in mothur was used for calculation of rarefaction curves, and the randomly sub-sampled taxonomic sequences (rarefaction depth = 13,800 sequences) were further used for the calculation of alpha and beta diversities. The calculation of various alpha diversity indices for richness, evenness, and phylogenetic diversity was performed while using the Summary.single tool in mothur. The mean alpha diversity indices among different groups of samples were compared based on one-way analysis of variance (ANOVA) with *vegan* package in R [36]. Beta diversity was calculated based on thetayc/jclass calculator and Phylip distance matrix by Dist.shared tool in mothur. The heatmaps for similarities in microbial communities were generated while using the Heatmap.sim tool in mothur. The dendrograms for compositional similarities were generated using Tree.shared and Newick.display tools in mothur. Non-metric multi-dimensional scaling (NMDS) analysis and analysis of similarities (ANOSIM) were performed with *vegan* package in R. The four-way Venn diagram with unique and shared OTUs were generated using *VennDiagram* package in R [37]. For all tests of significance, the significant level of *p* value less than 0.05 was used in order to determine the significance in differences among sample groups.

## 3. Results

### 3.1. Distinct Abundances of Skin Microbial Phyla

Various patterns of the microbial relative abundance on forearm skin at phylum level were observed in different groups of individuals (Figure 1). Though in general the skin microbial ecosystem was dominated by four major phyla, including Actinobacteria, Bacteroidetes, Firmicutes, and Proteobacteria, their relative abundances on the forearm skin were varied with farm workers’ animal operations and the types of animals. More specifically, the relative abundances of Actinobacteria, Bacteroidetes, Firmicutes, and Proteobacteria in forearm skin samples that were collected from individuals in Clarksville farm, where only dairy cattle are present, were 27.18, 4.77, 27.65, and 37.28%, respectively (Figure 1A). The relative abundances of Actinobacteria, Bacteroidetes, Firmicutes, and Proteobacteria in skin swabs that were collected from Upper Marlboro farm, which is an integrated farm and workers are exposed to animals, including beef cattle, sheep, chicken, and duck were 5.18, 2.74, 11.61, and 77.44% (Figure 1B), respectively. The relative abundances of Actinobacteria, Bacteroidetes, Firmicutes, and Proteobacteria in skin swab samples that were collected from Washington DC integrated farm, another integrated farm where the farm labors have operations on a variety of farm animals including cow, sheep, chicken, turkey, and pig, were 13.80, 3.63, 31.64, and 33.70% (Figure 1C), respectively.

As a control reference, the common phylum composition of forearm skin microflora in employees on College Park campus from the University of Maryland was also analyzed, and the Actinobacteria, Bacteroidetes, Firmicutes, and Proteobacteria pattern, of which displayed as 42.50, 5.09, 23.94, and 25.57%, respectively (Figure 1D). Overall, the relative epidermal abundances of Actinobacteria and Bacteroidetes in individuals with farm animal operations were significantly (*p* < 0.05) lower than that in the reference control, with the lowest level of all these three phyla being observed in Upper Marlboro that was integrated farm workers. Whereas, the relative epidermal abundances of Proteobacteria in individuals with farm animal operations were significantly (*p* < 0.05) higher than that in the reference control, with the highest level found in the same group.

### 3.2. Unique Skin Microbial Genera Compositions

The forearm skin microbiome exhibited a distinctive microbial relative abundance at the genus level in skin swab samples collected from each group of workers from various farm environments. In general, the top 35 most abundant genera and their relative abundances in skin swab samples that were collected from individual group were compared in this study (Figure 2). Overall, we found a relatively even distribution of forearm skin bacterial genera in the farm workers from Clarksville dairy farm and Washington DC integrated farm, partially reflected by their higher abundance of those genera other than the top ones (Others). Among the abundant bacterial genera, *Acinetobacter*, *Bacillus*, *Massilia*, *Pantoea*, *Pseudomonas*, *Rhizobium*, *Sphingomonas*, and *Stenotrophomonas*, were significantly (*p* < 0.05) higher in the farm workers, who were exposed to various animals, in comparison with the control group of individuals who did not have any close contact with farm animals at least 10 days prior to the skin swab collection. On the contrary, the relative abundances of *Anaerococcus*, *Aquabacterium*, *Chryseobacterium*, *Corynebacterium*, *Dietzia*, *Enterobacter*, *Janibacter*, *Lactobacillus*, *Micrococcus*, *Prevotella*, *Propionibacterium*, *Saccharibacteria*, *Staphylococcus*, *Streptococcus*, and *Veillonella* were significantly (*p* < 0.05) lower in the forearm skin samples that were collected from farm workers with frequent animal operations, as compared to those individuals in the reference control without farm animal exposure. Moreover, the relative abundances of the rest genera were found with irregularity among the groups.

The common and previously well-studied skin microbes, including *Staphylococcus*, *Streptococcus*, *Propionibacterium*, and *Corynebacterium,* were also detected in the samples that were collected from the Upper Marlboro integrated farm workers, with cow, sheep, chicken, and duck exposures (Figure 3). They exhibited the lowest levels as compared with the other groups, whereas their *Pseudomonas* and *Acinetobacter* were presented at the highest levels. Moreover, the skin microflora of Clarksville dairy farm workers with dairy cow operations share the similar lowest abundance of *Streptococcus* with those from workers in Upper Marlboro integrated farm, the highest abundance of *Corynebacterium* with those from individuals in reference control without farm animal contact, and moderate levels of *Staphylococcus*, *Pseudomonas*, *Propionibacterium*, and *Acinetobacter* with those from workers in Washington DC integrated farm with various farm animal operations.

### 3.3. Phylogenetic Distance and Relativeness of Skin Microbes

The microbial phylogenetic relativeness in forearm skin swab samples was analyzed and displayed by both dendrogram (Figure 4) and heatmap (Appendix A). Based on the dendrogram from Newick Display, the skin microbiome from various samples was categorized into three major clusters, which were in accordance with the groups and types of the sample sources: Cluster I, as the forearm skin microbiome of farm labors working in Clarksville dairy farm, Cluster II, as the forearm skin microbiome of the reference control from campus employees, and Cluster III, as the forearm skin microbiome of farm labors working in integrated farms (Upper Marlboro and Washington DC). Only one exception was found in skin sample #1, the donor of which is both dairy farm manager and campus employee, and it was classified into Cluster II (Figure 4). Furthermore, the heatmap of phylogenetic distance also illustrated that the forearm skin microbial species diversity was much phylogenetically related (higher similarity index, brighter redness) among samples that were categorized in the same cluster while phylogenetically distant (lower similarity index, darker redness) among samples from different clusters (Appendix A). Additionally, Figure 5 displays the commonness of skin microbial species among different combinations of groups. A total of 7583 microbial species were identified in all skin swab samples, in which they were unorthodoxly distributed and shared across different groups. The forearm skin samples that were collected from the individuals in Upper Marlboro integrated farm, Washington DC integrated farm, and the reference group all shared the most microbial species with those that were collected from the workers in Clarksville dairy farm, as 58.24, 46.58, and 57.68%, respectively, in terms of the ratio of commonness.

### 3.4. Dissimilarities in the Clustered Skin Microbial Compositions

Figure 6 shows the NMDS based on Bray–Curtis distances among grouped forearm skin microbial taxa. Through ANOSIM, we observed the overall significant (R^2^ = 0.641, *p* < 0.001) dissimilarity across different groups of samples. Pairwise Permutational Multivariate Analysis of Variance (PERMANOVA) indicated that insignificant (F = 2.171, R^2^ = 0.178, *p* = 0.156) difference was detected in the microbial composition of forearm skin microflora species between labors working in Clarksville dairy farm, which only contains dairy cows, and Washington DC integrated farm, which contains various farm animals, including dairy cows, sheep, chicken, turkey, and pigs. Whereas, significant (pairwise PERMANOVA with *p* < 0.01) dissimilarities were found with comparison between all other combinations.

### 3.5. Individual Species Diversity in Forearm Skin Microbiome

The alpha diversity indices, including invsimpson measuring phylogenic diversity, chao assessing microbial richness, and shannoneven evaluating microbial evenness, in skin microbiome were analyzed based on a rarefaction depth at 13,800 sequences, and they were significantly different in the skin swab samples that were collected from various farm worker groups (Figure 7). It was found that the invsimpson indices in forearm skin swab samples that were collected from individuals in Clarksville dairy farm were diverse in a range of 5.266–63.857, which was the highest average value (32.634) as compared with the swab samples that were collected from other farm worker groups as well as the control group. The lowest average invsimpson index (12.864) was observed in the forearm skin samples that were collected from reference control group. The average chao index values of skin swab samples were observed as 7807.892, 3940.490, 3449.457, and 3400.452 in the workers of Washington DC integrated farm, Clarksville dairy farm, Upper Marlboro integrated farm, and reference control group, respectively (Figure 7). The average shannoneven indices of the skin microflora were: 0.636, 0.593, and 0.489, and 0.448 in the workers of Clarksville dairy farm, Washington DC integrated farm, reference control group, and Upper Marlboro integrated farm, respectively. In addition, the total number of sequences in the skin microflora was also observed to be various, with the highest average value found in workers from Upper Marlboro integrated farm (71,671), followed by individuals in reference control group (47,336), workers from Clarksville dairy farm (40,785), and the lowest in workers from Washington DC integrated farm (36,464). Appendix A summarizes multiple other indices predicting alpha microbial diversity among different groups of skin samples, which were following similar patterns.

## 4. Discussion

Human skin is colonized by various microbes, predominantly bacteria, in more than 19 phyla, and the dominant phyla of the skin microbial community on healthy individuals consist of approximately 51.8% Actinobacteria, 6.3% Bacteroidetes, 24.4% Firmicutes, and 16.5% Proteobacteria [4,38,39]. We have detected that the forearm skin microflora of the individuals in the our reference control group was also dominated by these four phyla, and their compositions were 42.50% Actinobacteria, 5.09% Bacteroidetes, 23.94% Firmicutes, and 25.57% Proteobacteria, which indicated the considerable similarity with the previous studies and verified the methodology performed, the representability of the collected specimens, and appropriation of the reference control group. The phylum level abundances of skin microbes of the farm workers were noticeably altered by the concurrent operations on the respective farm animals. However, farm environments harboring different animal elements have established distinctive microbiome communities on the farm workers’ forearm skin, which demonstrated the discrete influences by individual type of livestock or poultry [9].

*Propionibacterium*, especially *P. acnes* and *Corynebacterium* as skin commensal bacteria in the phylum of Actinobacteria, are prevalent in sebaceous and moist sites, and researchers found that they play important roles as skin probiotics against pathogenic bacterial colonization and infections, such as MRSA and *Mycobacterium* [40,41]. Further, several genera of Proteobacteria, including *Pseudomonas* and *Acinetobacter*, are long-term epidermal residents, and they are involved in opportunistic pathogenic skin infections of immune-deficient individuals [42,43]. Here, in this study, we noticed that the relative abundances of epidermal *Propionibacterium* and *Corynebacterium* were substantially reduced in farm workers, especially those working in integrated farms. On the contrary, we also observed that the forearm skin microbiota in farm workers, especially integrated farm labors, were associated with the higher abundances of *Pseudomonas* and *Acinetobacter*. These results indicate that frequent operations on various livestock and poultry in individuals might shift their skin microbiome with less *Propionibacterium* and *Corynebacterium* and more *Pseudomonas* and *Acinetobacter*, which further escalates the risk of potential epidermal infections.

In general, *Staphylococcus* and *Streptococcus* are the most dominant and important Firmicutes in human skin flora and their functions were well defined as commensal [5]. For example, *S. epidermidis* and *S. mitis* are frequent and mutualistic residents of normal skin flora that promote immunity and protect skin from microbial infections [40,44]. In contrast, both *S. aureus* and *S. pyogenes* are also recognized as an opportunistic pathogens and they are able to cause skin wound infections, septicemia, and other invasive infections [40,44]. In this study, relatively less *Staphylococcus* and *Streptococcus* were detected in the forearm skin microflora of farm workers with frequent animal operations, while analyses of their specific species and associated functions are necessary for future studies that are based on a larger sample size. As a matter of fact, MRSA epidermal colonization has been previously reported in farmers with high-density livestock operations [45,46], which is connected with their antibiotic-resistant infections [21,47].

The symbiotic status of the skin microbial ecosystem is just like in the gastrointestinal system, which serves as the first defense and resistance toward infection and inflammation [48]. However, the perturbation and dysbiosis in skin microbiota with disrupted microbial biodiversity have been linked with multiple epidermal illnesses, including acne, dermatitis, eczema, folliculitis, psoriasis, and even skin cancers [49]. When compared with the normal forearm skin microbes of the individuals from the reference group in this study, in terms of the phylogenetic distance, richness, and evenness [50], frequent farm animal operations by the workers in the other groups have varied their biodiversity in the skin microflora. Although the forearm skin microbial communities in these farm workers were associated with relatively higher richness, they were observed with either lower evenness (workers in Upper Marlboro integrated farm) or higher phylogenetic distance (workers in Clarksville dairy farm and Washington DC integrated farm). These findings reveal that livestock or poultry could introduce extra microbial species in farm workers’ skin microflora and, at the same time, they also expand divergent lineages and disturb the original even distribution of the skin microbes [51,52].

Farm animal contact serves as the crucial factor in the alteration of skin microbiome in the farm workers investigated in the current study. Simultaneously, different types of farm animals, possessing distinctive microbiota compositions, synergistically contribute to the divergent forearm skin microbial ecosystem [1,10]. For instance, the skin of non-human mammals, especially cattle, sheep, and pigs, is colonized by a predominantly lower number of Actinobacteria, but higher Proteobacteria than those on human beings [10], while relatively lower abundances of both Actinobacteria and Bacteroidetes were associated with poultry carcass [53]. These animals are all present in the integrated farms investigated in this study, which might partially explain the unique epidermal microbial composition that formed in the workers from the Upper Marlboro integrated farm. In the meantime, Verdier-Metz et al. detected the substantial dominance of Firmicutes with limited numbers of Actinobacteria and Bacteroidetes in dairy cow teat skin microflora [54]; moreover, several core genera in raw milk include *Acinetobacter*, *Corynebacterium*, and *Staphylococcus* [55]; all of these are in support of the development of distinctive microbial compositions in the workers from Clarksville and Washington DC farms, where they had frequent dairy cattle operations and close contact with their milk products. Additionally, the influence of dairy cattle operations might explain the similarity by NMDS that was found in the clustered skin microbial compositions between workers from Clarksville dairy farm and Washington DC integrated farm as well.

Even in this study, we identified distinctive microbial compositional patterns on skin of workers in farm with different animal commodities and notified the possibility of epidermal infections that were caused by zoonotic pathogens; it has several limitations that lead to further in-depth study in the future. First, the sample size of human forearm skin is relatively small, for which we obtained a relatively high variations in the sequence numbers. Specifically, less samples were collected from the Washington DC integrated farm, due to the unwillingness of participation of their farm workers. Subsequently, this study eliminated the impacts of antibiotic treatment and pet contact on skin microbiome, whereas it is difficult to exclude or control the influence of using hand sanitizers. Therefore, the impact of hand hygiene products on forearm skin microbiota could be separately investigated in the future. Further, culture-based morphological and biochemical identification or whole-genome sequencing could be further performed in order to discriminate staphylococci at the species level, which might provide more information about its epidermal transmission towards farm workers by means of direct contact within farm animals.

## 5. Conclusions

Reduced levels of *Staphylococcus* and *Streptococcus* in forearm skin microbiome were associated with interaction between farm workers and animals. Significantly lower ratios of epidermal Actinobacteria, especially *Corynebacterium* and *Propionibacterium*, as well as higher ratios of epidermal Proteobacteria, especially *Pseudomonas* and *Acinetobacter*, were observed in integrated farm workers, where they frequently worked with various types of farm animals. Whereas, frequent dairy cow operations introduced a higher abundance of Firmicutes to the skin microbiota of the farm workers. The environmental influences of agricultural livestock and poultry operations on modulating the forearm skin microbiome of the on-site farm workers were also reflected by the enriched microbial species, but disturbed evenness and phylogenetic relativeness, which may further increase the risk of developing skin diseases, particularly infections that are caused by multi-drug resistant bacterial pathogens.

## Figures and Tables

**Figure 1 microorganisms-08-01481-f001:**
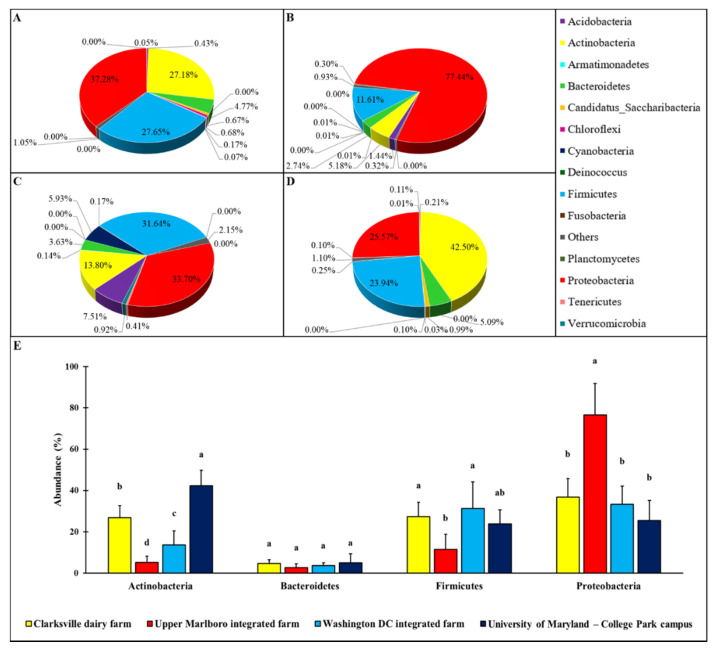
Relative percentage of the phylum level abundance in forearm skin microflora. The relative skin microbial abundance in the investigated individuals from the dairy farm in Clarksville (**A**), the integrated farm in Upper Marlboro (**B**) and Washington DC (**C**), and University of Maryland—College Park campus (**D**) was analyzed and compared at the phylum level. The top four phyla across each group are shown in (**E**). The a, b, c, and d indicate significant differences based on ANCOM.

**Figure 2 microorganisms-08-01481-f002:**
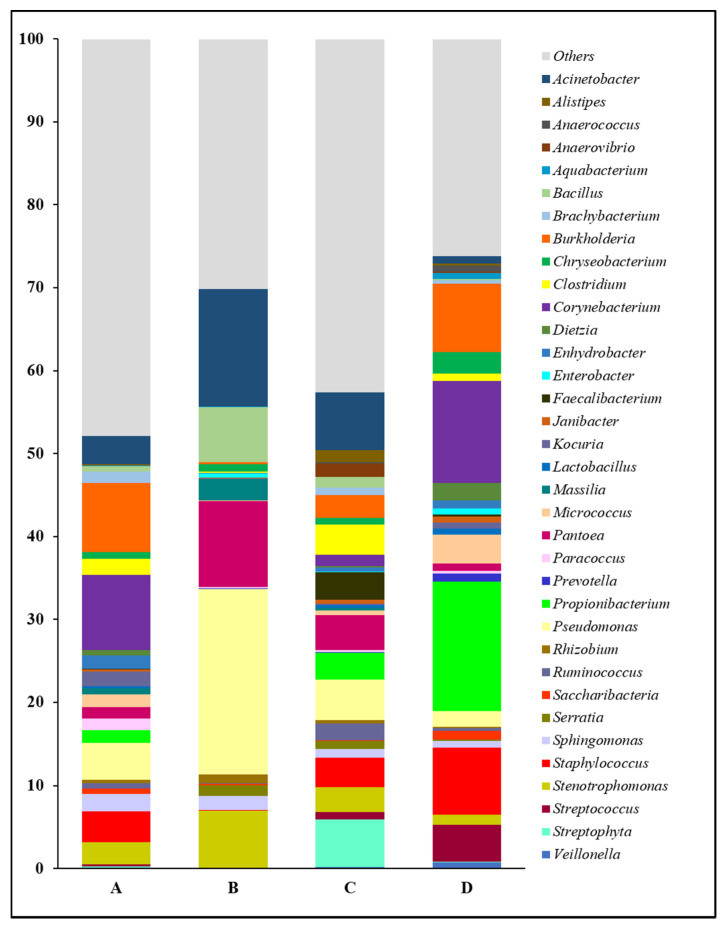
Relative abundance of forearm skin microflora at genus level. The relative skin microbial abundance of the top 35 genera in the investigated individuals from the dairy farm in Clarksville (**A**), the integrated farm in Upper Marlboro (**B**) and Washington DC (**C**), and University of Maryland—College Park campus (**D**) was analyzed and compared.

**Figure 3 microorganisms-08-01481-f003:**
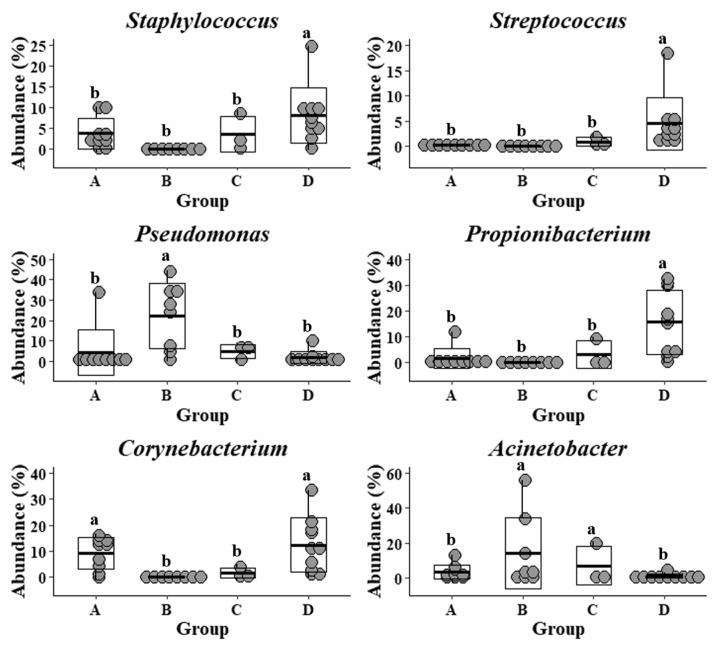
Relative abundance of specific genera in forearm skin microflora. The relative skin microbial abundance of *Staphylococcus*, *Streptococcus*, *Pseudomonas*, *Propionibacterium*, *Corynebacterium*, and *Acinetobacter* in the investigated individuals from the dairy farm in Clarksville (**A**), the integrated farm in Upper Marlboro (**B**) and Washington DC (**C**), and University of Maryland—College Park campus (**D**) was analyzed and compared. Letters a and b indicate significant differences among groups that are based on ANCOM.

**Figure 4 microorganisms-08-01481-f004:**
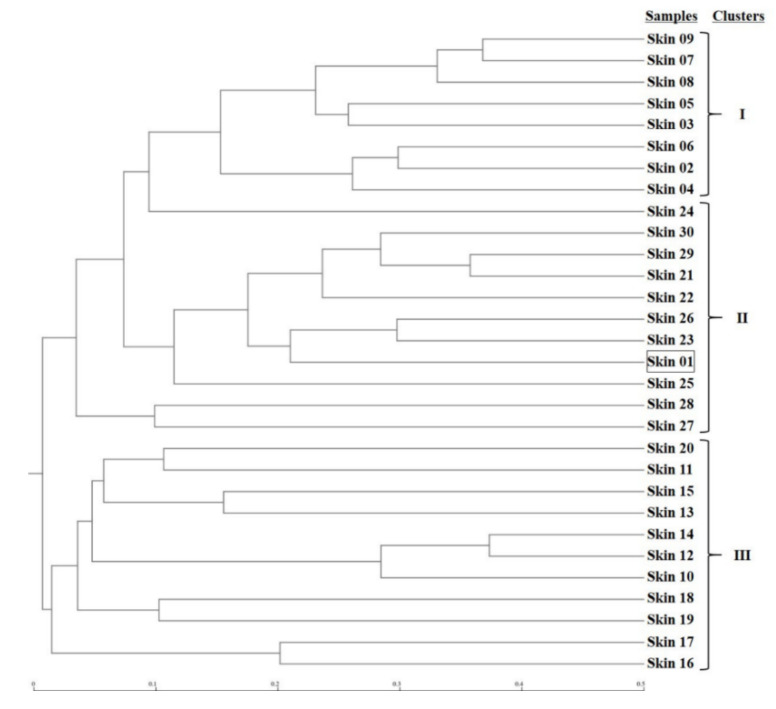
Dendrogram for phylogenetic distance of forearm skin microbiome. Clusters I, II, and III were predicted and classified among forearm skin microflora from various sample sources based on Newick Standard. The square represents skin sample #1 as an outlier. The scale bar unit represents 0.1 Phylip distance index value between the two samples.

**Figure 5 microorganisms-08-01481-f005:**
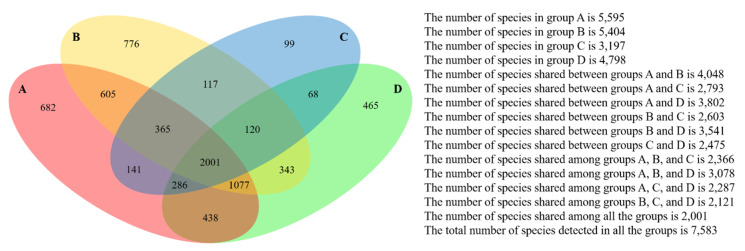
Commonness of forearm skin microbes across different groups. Four-way Venn diagram based on level 0.03 (97% similarity at species level) encloses all 30 datasets categorized in four groups, including Group A—Clarksville dairy farm, Group B—Upper Marlboro integrated farm, Group C—Washington DC integrated farm, and Group D—University of Maryland—College Park campus.

**Figure 6 microorganisms-08-01481-f006:**
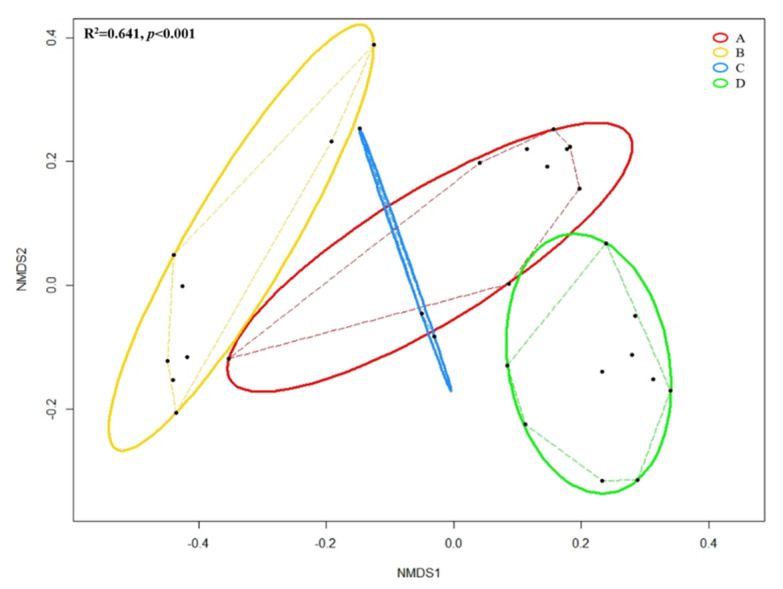
Non-metric multidimensional scaling (NMDS) based on Bray–Cutis distance matrix enclosing all 30 datasets. Black dot represents individual skin microbial community; Solid lines represent ellipses enclosing all points in different groups; Dashed lines represent convex hulls by ordihull function. Significance across groups (*p*-value) based on ANOSIM; Group A: Clarksville dairy farm; Group B: Upper Marlboro integrated farm; Group C: Washington DC integrated farm; and, Group D: University of Maryland—College Park campus.

**Figure 7 microorganisms-08-01481-f007:**
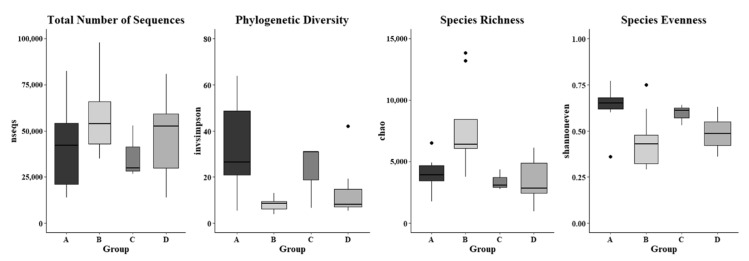
Microbial diversity of forearm skin microbial ecosystem. Alpha diversity indices of nseqs (total number of sequences), invsimpson (phylogenic diversity), chao (species richness), and shannoneven (species evenness) in the investigated individuals from the Dairy farm in Clarksville (**A**), the integrated farm in Upper Marlboro (**B**) and Washington DC (**C**), and University of Maryland—College Park campus (**D**). Letters a and b indicate significant differences among groups based on ANOVA. The widths of the boxes represent the variances in the observations.

**Table 1 microorganisms-08-01481-t001:** Sample sources and numbers.

Group	Location	Category	Number	Farm Commodity
A	Clarksville	Dairy Farm	9 (Skin1–9)	Dairy Cattle
B	Upper Marlboro	Integrated Farm	8 (Skin10–17)	Beef Cattle, Sheep, Chicken, Duck, Produce
C	Washington DC		3 (Skin18–20)	Dairy Cattle, Sheep, Chicken, Turkey, Pig, Produce
D	College Park	Control	10 (Skin21–30)	-

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
