# Peer review of "Environmental Influences of High-Density Agricultural Animal Operation on Human Forearm Skin Microflora"

_microorganisms, 2020, doi:10.3390/microorganisms8101481_

Round 1

Reviewer 1 Report

Peng and Biswas present in their work the descriptive comparison within 3 different forearm skin microbiomes of farm workers and a distinct control group. Although the formulated question and results are interesting, the analysis shows deficits in study design, statistics, and visualization. Please find my remarks and comments below:

Group C contains only 3 samples in comparison to A, B, and D. Could the authors explain this bias? In addition, later in the manuscript one sample of the control group is described as outlier because of association to the study groups. This sample should be excluded from the whole analysis as it can skew the result.

line 97: Antibiotic treatment within the last 10 days is quite a short timespan. Especially because inclusion criteria for the control group (line 100: Campus staff antibiotic treatment within the last 30 days.) is 30 days without AB treatment. Authors should explain why the inclusion criteria in respect to antibiotic treatment differences between the two groups.

In addition I would be interested if members of the control group house pets.

line 134: add used mothur version
line 141-142: version of SILVA database and RDP classifier version are missing.

Authors state that samples were collected in duplicates. Only 30 samples are publicly available on SRA in contrast to the expected 60. Please clarify what happened to the duplicates and how they were treated / merged for further analysis and processing.

line 146 (and following) References for R, RStudio if used and r-packages are missing.

Figure 1: Illustration unusual and confusing. Boxplots or stacked bar charts will make a more clear point here. Whereby pooled group representations can be quite easily biased by one single sample, hence I would recommend boxplots.

line 151: Please add rarefaction depth.

As the forearm skin microbiome is quite a common location of 16S metagenomic surveys results should be also discussed in regard to already available datasets to show representability of the used specimens.

Metadata about skin care products is missing as I would assume that farm workers take special care as their forearm skin is highly stressed. Maybe also sanitizers are used. This would have a crucial impact on the analysis result.

Figure2: In case only top 35 phyla are shown, remaining should be summarized to eg "Other"  and included into the illustration. Otherwise the plot can be easily misinterpreted.

Figure 3: Dot size should be decreased to increase readability. Whiskers should be included into the boxplot representations. Representation of sig. differences between groups is quite confusing. Please clarify and improve the presentation in plot.

line 267: Please add rarefaction depth at which the indices have been calculated.

figure 5: Very crowded figure and text is hard / impossible to read. Please improve.

figure 5b: Variation of number of sequences between groups is quite high. Please discuss. Why is the box of group c not as wide as the others?

line 294: Pairwise comparison after a non significant group comparison is statistically not valid.

line 288: Authors should explain why Bray-curtis distance was used instead of UniFrac or weighted UniFrac.

Author Response

Group C contains only 3 samples in comparison to A, B, and D. Could the authors explain this bias? In addition, later in the manuscript one sample of the control group is described as outlier because of association to the study groups. This sample should be excluded from the whole analysis as it can skew the result.

Response: We appreciate reviewer’s comment. Due to unwillingness of a large number of the farmworkers in group C in participating in this skin microbial study, we had limited number of samples from that farm in comparison with the other farms in groups A, B, and D. For the outliers in the dendrogram, the skin microbiota of Sample #1 was clustered with those of the controls based on only phylogenetic distance. We believe that the phylogenetic distance outlier still contains valuable information about microbial abundances as well as community richness and evenness, since similar significant results were obtained from the analyses excluding Sample #1.

line 97: Antibiotic treatment within the last 10 days is quite a short timespan. Especially because inclusion criteria for the control group (line 100: Campus staff antibiotic treatment within the last 30 days.) is 30 days without AB treatment. Authors should explain why the inclusion criteria in respect to antibiotic treatment differences between the two groups. In addition I would be interested if members of the control group house pets.

Response: Thank you for pointing these out. The ‘10 days’ at line 97 was a typo. We have the same criterion of antibiotic treatment timespan for inclusion and exclusion, which is ‘30 days’. For one of the inclusive criteria of control individuals, we required ‘no personal contact with farm animals and pets within the recent 10 days’, which was not clearly addressed before, but it has been clarified now at lines 101-102.

line 134: add used mothur version.

Response: We would like to thank reviewer for pointing out this error. Now we have added the version number of mother used for data analysis at line 135.

line 141-142: version of SILVA database and RDP classifier version are missing.

Response: In response to the reviewer’s comment, we have added the version numbers of SILVA database and RDP classifier used for data analysis at line 143.

Authors state that samples were collected in duplicates. Only 30 samples are publicly available on SRA in contrast to the expected 60. Please clarify what happened to the duplicates and how they were treated / merged for further analysis and processing.

Response: We appreciate the reviewer’s time for going through our manuscript in details. For clarification, description of the details about processing sample duplicates has now been added in the Method section at lines 105-106 and 112-113. For sample collection, we collected duplicated swabs from each farmworker. The DNA of duplicated swab samples for each individual were combined and recognized as one DNA sample for further sequencing and analysis.

line 146 (and following) References for R, RStudio if used and r-packages are missing.

Response: We would like to thank you for pointing this out. We have inserted corresponding references for the R-packages at lines 152, 157, and 163.

Figure 1: Illustration unusual and confusing. Boxplots or stacked bar charts will make a more clear point here. Whereby pooled group representations can be quite easily biased by one single sample, hence I would recommend boxplots.

Response: We thank the reviewer for giving such good suggestion. We have added a separate graph in addition to the pie chart, specifically illustrating the top 4 phyla in the forearm skin microbiota (Figure 1).

line 151: Please add rarefaction depth. line 267: Please add rarefaction depth at which the indices have been calculated.

Response: Thank the reviewer for providing such good suggestion. The rarefaction depth has been described both in the Method section (line 153) and Result section (line 288).

As the forearm skin microbiome is quite a common location of 16S metagenomic surveys results should be also discussed in regard to already available datasets to show representability of the used specimens.

Response: In the Discussion section, we compared the forearm skin microbial community composition obtained from this study with the findings from previously published references. The considerable similarity could indicate the representability of the specimens used in this study (lines 314-321).

Metadata about skin care products is missing as I would assume that farm workers take special care as their forearm skin is highly stressed. Maybe also sanitizers are used. This would have a crucial impact on the analysis result.

Response: We thank the reviewer’s critical comment. The impact of skin care products on forearm skin microbiome could be another interesting research study (line 387-390). However, for the current study, the farm workers were not willing to inform us any information about the types of sanitizers they used at home. However, the changes of forearm skin microbiota in the workers were mainly induced by long-time and frequent farm animal contacts during work time every day.

Figure2: In case only top 35 phyla are shown, remaining should be summarized to eg "Other" and included into the illustration. Otherwise the plot can be easily misinterpreted.

Response: We thank the reviewer for the suggestion. We have now displayed the abundance of other phyla in a new group as ‘Others’ in Figure 2. The corresponding content of illustration has also been added at lines 202-205.

Figure 3: Dot size should be decreased to increase readability. Whiskers should be included into the boxplot representations. Representation of sig. differences between groups is quite confusing. Please clarify and improve the presentation in plot.

Response: In accordance with the reviewer’s comments, we have slightly decreased the dot size since we aimed to highlight each individual dots, and the whiskers for boxplot have also been added. The representation of significant differences has now been indicated by significance letters in Figure 3.

figure 5: Very crowded figure and text is hard / impossible to read. Please improve.

Response: We thank the suggestion from the reviewer. In accordance with the comment, we have separated the original Figure 5 into three individual figures (Figure 5a→Figure 5; Figure 5b→Figure 7; Figure 5c→Figure 6). The corresponding figure numbers in text have also been changed throughout the Result section.

figure 5b: Variation of number of sequences between groups is quite high. Please discuss. Why is the box of group c not as wide as the others?

Response: Due to the variations in skin microbiota of individuals, the variation bars of sequence numbers across groups were relatively high (lines 384-385). The boxes were drawn with widths proportional to the square-roots of the number of observations in the groups (line 312). Less width in Group C means less variance in the observations.

line 294: Pairwise comparison after a non significant group comparison is statistically not valid.

Response: The overall comparison indicated significant (R2=0.641, p<0.001) dissimilarity of microbial composition among all four groups. Then, pairwise comparisons were performed between each two groups for potential significance identification. We have clarified this at line 271 by indicating ‘pairwise’.

line 288: Authors should explain why Bray-curtis distance was used instead of UniFrac or weighted UniFrac.

Response: Bray-Curtis distance is used to quantify the compositional dissimilarity in ecological studies. A great number of microbiome research performed Non metric multidimensional scaling beta diversity analysis based on Bray-Curtis distance matrix. Although UniFrac is more phylogenetically aware, it treats the closely related microbes as more identical. For our study, we aimed to identify the compositional dissimilarity of forearm skin microbiota among different groups of individuals, for which we hoped to avoid treating phylogenetically close-related skin microbes as similar or identical. Therefore, Bray-Curtis distance was selected for analysis.

Reviewer 2 Report

This compact, detailed study on a small sample can be considered as a trial stage of a large study, performed to work out the protocol or research pipeline.

Several remarks

The authors focused their attention on the discussion of widely known taxa, however, other representatives of Proteobacteria (f.e., Stenotrophomonas, Burkholderia etc.) are worthy of careful consideration.

With regard to staphylococci, of course, a greater number of species of both coagulase positive and coagulase negative staphylococci should be analyzed, but this is the groundwork for research with a different methodological approach that allows to determine the species of microorganism.

Author Response

The authors focused their attention on the discussion of widely known taxa, however, other representatives of Proteobacteria (f.e., Stenotrophomonas, Burkholderia etc.) are worthy of careful consideration.

With regard to staphylococci, of course, a greater number of species of both coagulase positive and coagulase negative staphylococci should be analyzed, but this is the groundwork for research with a different methodological approach that allows to determine the species of microorganism.

Response: We thank the reviewer for recognizing the importance of our study. Regarding the comments on staphylococci, we agree that species-level identification is important, which will be targeted in our future work. The limitation of this study has been addressed at the end of the Discussion section (lines 390-393).

Round 2

Reviewer 1 Report

Thx to the authors for discussing my comments and questions and their efforts towards improving the quality of presented figures. I'd like to point out the importance of metadata in metagenomic surveys. Hence, I recommend getting consent on participating and sharing metadata already before study start to overcome issues of sample number bias in investigated groups and the resulting effects.